# Evaluation of Aerogel Spheres Derived from *Salix psammophila* in Removal of Heavy Metal Ions in Aqueous Solution

**Yuan Zhong [1], Yuhong An [1], Kebing Wang [2], Wanqi Zhang [1], Zichu Hu [2], Zhangjing Chen [3], Sunguo Wang [4], Boyun Wang [2], Xiao Wang [2], Xinran Li [5], Xiaotao Zhang [2,6,*] and Ximing Wang [1,6]**

[1] College of Material Science and Art Design, Inner Mongolia Agricultural University, Hohhot 010018, China; zhongyuan@emails.imau.edu.cn (Y.Z.); anyuhong@emails.imau.edu.cn (Y.A.); nmgndcyyzwq@emails.imau.edu.cn (W.Z.); wangximing@imau.edu.cn (X.W.)

[2] College of Science, Inner Mongolia Agricultural University, Hohhot 010018, China; wkb6658@gmail.com (K.W.); hzc101@emails.imau.edu.cn (Z.H.); 13485473614@emails.imau.edu.cn (B.W.); 1491831205@emails.imau.edu.cn (X.W.)

[3] Department of Sustainable Biomaterials, Virginia Polytechnic Institute and State University, Blacksburg, VA 24060, USA; chengo@vt.edu

[4] Sungro Bioresource & Bioenergy Technologies Corp., Edmonton, AL T6R3J6, Canada; wangsunguo@gmail.com

[5] College of Chemistry & Environment, Southwest Minzu University, Chengdu 610041, China; lixinran0903@emails.imau.edu.cn

[6] Inner Mongolia Key Laboratory of Sandy Shrubs Fibrosis and Energy Development and Utilization, Hohhot 010018, China

\* Correspondence: xiaotaozhang@imau.edu.cn

**Abstract:** Heavy metal wastewater treatment is a huge problem facing human beings, and the application degree of *Salix psammophila* resources produced by flat stubble is low. Therefore, it is very important to develop high-value products of *Salix psammophila* resources and apply them in the removal heavy metal from effluent. In this work, we extracted the cellulose from *Salix psammophila*, and cellulose nanofibers (CNFs) were prepared through TEMPO oxidation/ultrasound. The aerogel spheres derived from *Salix psammophila* (ASSP) were prepared with the hanging drop method. The experimental results showed that the Cu(II) adsorption capacity of the ASSP composite (267.64 mg/g) doped with TOCNF was significantly higher than that of pure cellulose aerogel spheres (52.75 mg/g). The presence of carboxyl and hydroxyl groups in ASSP enhanced the adsorption capacity of heavy metals. ASSP is an excellent heavy metal adsorbent, and its maximum adsorption values for Cu(II), Mn(II), and Zn(II) were found to be 272.69, 253.25, and 143.00 mg/g, respectively. The abandoned sand shrub resource of SP was used to adsorb heavy metals from effluent, which provides an important reference value for the development of forestry in this sandy area and will have a great application potential in the fields of the adsorption of heavy metals in soil and antibiotics in water.

**Keywords:** sandy shrub; cellulose; TEMPO; aerogel; adsorption

## 1. Introduction

*Salix psammophila* is one of the characteristic sandy shrubs in North and Northwest China [1,2]. *Salix psammophila* is characterized by simple reproduction, quick growth, high resistance to barren drought, and stubble rejuvenation [3,4]. *Salix psammophila* is mainly composed of cellulose, hemicellulose, lignin, and a small amount of ash [5]. Cellulose is its main component and can be extracted through certain treatments and modified by sodium periodate, 2,2,6,6-tetramethylpiperidine-1-oxyl (TEMPO) [6,7] and other oxidation systems [8] to obtain different functional groups. Then, the modified cellulose can be treated with acid [9], mechanical treatment [10,11] and bacterial degradation [12] to prepare finely structured cellulose with a diameter of 1–100 nm, also called nanocellulose [13]. Li Mei [14] used recycled polypropylene and *Salix psammophila* powder as the main raw

materials and prepared wood–plastic composites with a molding method. Xiao Liu [15] prepared activated carbon-based *Salix psammophila* with phosphoric acid activation, and the maximum adsorption capacity of the sulfamethazine sodium was found to be 338.58 mg/g. However, there have been few reports on *Salix psammophila* cellulose adsorbents. Therefore, it is of great significance to expand the utilization field and improve the utilization value of the *Salix psammophila* resources produced by flat stubble [16].

An aerogel sphere is a kind of porous solid material with high porosity, ultra-low density, and heat insulation that is composed of polymer molecules or colloidal particles intertwined into a nanoporous network structure and filled with air [17,18]. Cellulose aerogel spheres have become one of the most studied new aerogel materials [19,20]. Aerogel spheres derived from *Salix psammophila* (ASSP) present biodegradable, biocompatible, and renewable excellent features, and in the application of high-performance composite materials, they show great potential [21]. They are used for wastewater treatment to realize the concept of "using waste treat waste" and achieving a unity of social, economic, and environmental effects of profound significance.

Heavy metal ions are widely derived from wastewater discharge from industrial metallurgy, ore smelting, and electroplating. They exist in ionic forms, mainly Cu(II), Zn(II), Cr(III), and Mn(II) [22], and they spread to groundwater with precipitation, thus seriously endangering human health [23]. Amphol Daochalermwong [24] used cellulose fiber from pineapple leaves as a raw material and modified it with carboxymethyl groups to produce Cell-CM. Their experiments showed that the maximum adsorption capacities of Cell-CM for Pb(II) and Cd(II) were 63.4 and 23.0 mg/g, respectively. However, Cell-CM is inconvenient to recycle. Sami Guiza [25] used cellulosic waste orange peel (CWOP) to remove Cu(II) from water. The CWOP maximum adsorption capacity was found to be 63 mg/g, so the absorption performance of CWOP was slightly poor. It can be seen that biomass materials with easy recovery and high adsorption have great potential in heavy metal wastewater treatment.

Based on the previously reported extraction of *Salix psammophila* microcrystalline cellulose, and the authors of this work propose the preparation of aerogel spheres derived from *Salix psammophila* because the material has better heavy metal adsorption performance than traditional pure cellulose adsorbents. The *Salix psammophila* nanofibers were prepared through TEMPO oxidation combined with ultrasound. The aerogel spheres were prepared with the hanging drop method with *Salix psammophila* nanofibers and *Salix psammophila* microcrystalline cellulose solutions. With the aid of porous structures and carboxyl functional groups, the goal of efficiently removing metal ions from water was achieved.

The purpose of this work was twofold: to propose a type of waste biomass resource utilization direction through the application of *Salix psammophila* nanofibers and to provide a carrier idea for subsequent application in catalysis and adsorption materials through the adsorption properties of aerogel spheres derived from *Salix psammophila*.

## 2. Materials and Methods

### 2.1. Materials

Samples of *Salix psammophila* were collected from the Kabuki Desert of Ordos, China. Samples were cleaned with water, dried, and crushed; 100–120 mesh of powder were measured for reserve. We used the following materials: cuprizone from Tianjin Beilian Fine Chemicals Development Co., Ltd. (Tianjin, China); ascorbic acid, citric acid, anhydrous sodium acetate, xylenol orange, anhydrous ethanol, sodium oxide, sodium hypochlorite, and potassium iodide (effective chlorine content 10%) from Tianjin Kemao Chemical Reagent Co., Ltd. (Tianjin, China); potassium periodate, ammonia solution, $MnCl_2$, ZnO, $CuSO_4$, and urea from Tianjin FengChuan chemical reagent Technology Co., Ltd. (Tianjin, China); sodium pyrophosphate and potassium hydroxide from Tianjin Damao Chemical Reagent Factory (Tianjin China); glacial acetic acid and tert-butanol from Tianjin Fuchen Chemical Reagent Co., Ltd. (Tianjin, China); sodium chlorite from Aladdin Reagent Co., Ltd. (Beijing, China); and 2,2,6,6-tetramethylpiperidine-1-oxygen radical and sodium bro-

mide from McLean Co., Ltd. (Shanghai China); See below for the test method and standard curve (Figure S1) of heavy metals.

## 2.2. Preparation of Salix psammophila Microcrystalline Cellulose

The extraction of *Salix psammophila* microcrystalline cellulose was conducted according to the methods of previous papers [26]. Ten grams of *Salix psammophila* were weighed and heated in distilled water at 1:60 g/mL for 1 h at 80 °C. Impurities and some water-soluble components were removed by filtration, and the yield was 87%. The nitric acid–ethanol solution (1:3 *v/v*, 1:43 g/L) was weighed and placed in a three-necked flask, condensed, and refluxed at 90 °C. After the sample was removed, it was washed to neutral with distilled water and anhydrous ethanol, and then it was dried at 105 °C with constant weight. After drying, it was added to a 7.5 wt% sodium chlorite solution (1:20 g/mL), adjusted to a pH of 3–4 with glacial acetic acid, reacted at 75 °C for 3 h, washed with distilled water and anhydrous ethanol to neutral, and dried at 60 °C to a constant weight. The bleached samples were then placed in a 10% KOH (1:20 g/mL) solution, magnetized at 75 °C for 2 h, washed to neutral, and dried to a constant weight at 60 °C. Finally, 8 wt% hydrochloric acid solution (1:20 g/mL) was hydrolyzed at 90 °C for 1.5 h. The product was successively washed with distilled water and anhydrous ethanol to neutral. After drying, the sample was named SP-Mic-C.

## 2.3. Preparation of Salix psammophila Nanofibers

We added 1.000 g of SP-Mic-C, 0.032 g of TEMPO, 0.603 g of sodium bromide, and 100 mL of distilled water to a 250 mL three-neck flask. After stirring at 60 °C for 30 min, 20 mL of a sodium hypochlorite solution (10 wt%) were added, and a 0.5 M sodium hydroxide solution was used to adjust the pH of the solution to between 10.0 and 10.5 until the end of the reaction. Anhydrous ethanol (20 mL) was used to quench the reaction, and the oxidized SP-Mic-C solution was ultrasonicated for 2 s and rested for 3 s at 1200 W for 90 min. Under the weak base condition, the cellulose was fully moistened and expanded in the solution. After the reaction, the solution pH was adjusted to acidic with hydrochloric acid and then washed to neutral. After lyophilization, the sample was named *Salix psammophila* nanofiber (TOCNF), and the yield was about 71.51%.

## 2.4. Preparation of Salix psammophila Microcrystalline Cellulose/Salix psammophila Nanofiber Aerogel Spheres

We added 4.000 g of SP-Mic-C and 6.672 g of NaOH to 77.76 mL of distilled water. After stirring until the cellulose solution was homogeneous, we added 11.52 g of urea and stirred again until it was dissolved, refrigerated at −12.5 °C, and obtained a 4 wt% SP-Mic-C solution. We weighed an amount of SP-Mic-C solution (g) and 0.05 mg/mL of TOCNF (g), mixed them well, and we used a 5 mL syringe to drop the mixture into a coagulation bath of ice acetic acid, carbon tetrachloride, and ethyl acetate; after immersion in distilled water, anhydrous ethanol, and tert-butyl alcohol for 30 min, the soaked aerogel spheres were freeze-dried into aerogel spheres. The samples were named as follows: the quantities of *Salix psammophila* microcrystalline cellulose solution-mass of solution (g)-*Salix psammophila* nanofiber solution-mass of solution (g), i.e., SC-10-TOCNF-0 (None), SC-9.5-TOCNF-0.5, SC-9.0-TOCNF-1.0, SC-8.5-TOCNF-1.5, SC-8.0-TOCNF-2.0, and SC-7.5-TOCNF-2.5.

## 2.5. Heavy Metal Adsorption

We accurately weighed 0.0500 g of aerogels (BS210S, Sartorius, Gottingen, Germany) and added that to 50 mL of a solution with a known concentration. The suspension was stirred at a constant speed of 120 rpm at a constant temperature with a shaking table (Zhangjiagang sha-c, China), and its pH value was adjusted with the needed amount of hydrochloric acid solution or sodium hydroxide solution (sartorius PB-10, Germany). After reaching the adsorption equilibrium, the mixture was centrifuged at 10,000 rpm for 10 min (h2050r, Changsha, China). The residual concentration of heavy metals in the supernatant

was determined with a UV–Vis spectrophotometer [27] (TU-1901, Beijing, China). The adsorption experiments were carried out under different initial concentrations, pH values, adsorption temperatures, and times. Considering experimental error, the three experiments were run in parallel under the same conditions, and the results are reported as the average. The adsorption capacity of heavy metal ions was determined with the following equation:

$$q_e = \frac{(C_0 - C_e)V}{M} \tag{1}$$

where $q_e$ is the adsorption capacity of the sample, mg/g; $C_e$ is the equilibrium concentration of heavy metal ions, mg/L; $C_0$ is the initial concentration of the heavy metal ion solution, mg/L; V is the volume of the heavy metal ion solution, L; and M is the amount of adsorbent, g.

*2.6. Adsorption Kinetics*

To study the potential rate-controlling steps of adsorption, pseudo-first-order and pseudo-second-order kinetic models are generally used to fit the experimental data, as described by Equations (2) and (3), respectively:

$$\log(q_e - q_t) = \log q_e - \frac{K_1 t}{2.303} \tag{2}$$

$$\frac{t}{q_t} = \frac{1}{K_2 q_e^2} + \frac{t}{q_e} \tag{3}$$

where $q_e$ and $q_t$ are the amounts of heavy metal ions adsorbed (mg/g) at equilibrium and at time t (min), respectively; $K_1$ (min$^{-1}$) is the pseudo-first-order rate constant; $K_2$ g·(mg/min)$^{-1}$ is the rate constant of the pseudo-second-order adsorption kinetic equation; $\log(q_e - q_t)$, $t/q_t$ is the *y* axis; and t is the *x* axis, The data are linearly fitted, and $K_1$ and $K_2$ are obtained according to the slope and intercept.

*2.7. Characterization*

Scanning electron microscopy (SEM) (Hitachi 4800; Hitachi Limited, Tokyo, Japan) was used to observe the changes of the sample surface during the transformation from raw material to cellulose. The scanning electron microscope was operated at an accelerating voltage of 10 kV. A small number of samples were placed on conductive adhesive and then gold-sprayed (small ion sputtering instrument; Beijing Zhongke Instrument Co., Ltd., Beijing, China), and their morphology was observed via with the FEI Tecnai S-200kV twin field emission high-resolution transmission electron microscope (Thermo Fisher Scientific, Waltham, MA, USA). The sample was diluted with anhydrous ethanol, dried in copper mesh, and tested. The FTIR spectra were measured using a Perkin Elmer 65 spectrometer (Perkin Elmer Instruments Co., Ltd., Waltham, MA, USA). The experimental parameters included a range of 4000–600 cm$^{-1}$, a cumulative number of scans of 32, and a resolution of 1 cm$^{-1}$. The TGA experiments were performed with an HCT-1 integrated thermal analyzer (Beijing Hengjiu Scientific Instrument Factory, Beijing, China). The samples were analyzed in an N$_2$ atmosphere with a pressure of 0.3 MPa, a flow rate of 20 mL/min, a heating rate of 10 °C/min, and a temperature range from approximately 25 to 800 °C. The BETs of aerogel spheres were determined with an ASAP 2020HD88 static nitrogen adsorption apparatus. The samples were first vacuum-pretreated at 105 °C for 3 h to remove water and air, and then the specific surface area of the aerogel was determined in a liquid nitrogen environment.

**3. Results and Discussion**

*3.1. Characterization of the Salix psammophila Nanofibers*

Figure 1 shows SEM images of the transformation from SP-Mic-C to TOCNF. The SP-Mic-C (Figure 1a) still existed in the form of unopened fibril and had distinct particles

with diameters of about 4–50 μm. Following the cavitation effect of the high-intensity ultrasound, the generated cellulose nanofibers (Figure 1b) agglomerated and intertwined with each other to construct a 3D network structure. Abundant hydroxyl groups on the surface make it easier to form hydrogen bonds, and the interactions between hydrogen bonds between cellulose and van der Waals forces led to the formation of cellulose nanofiber aggregates [28]. Figure 1c shows TEM images of TOCNF. Due to the electrostatic repulsion generated by the carboxyl group, the nanocellulose generally presented a long whisker structure, though there was still a small amount of incomplete broken cellulose in the solution. Figure 1f shows the diameter distribution of TOCNF. The average diameter of TOCNF was found to be 23.39 nm. The FTIR of SP-Mic-C to TOCNF were analyzed, as shown in Figure 1d,e. The peak at 3330–3350 cm$^{-1}$ corresponded to an –O–H stretching vibration peak. After oxidation, the peak value increased due to the introduction of a carboxyl group. The stretching vibration peak of –C=O was near 1734 cm$^{-1}$. The asymmetric contraction vibration of carboxyl was near 1600 cm$^{-1}$ [29,30]. After TEMPO oxidation, the $C_6$ hydroxyl group was transformed into a carboxyl group, resulting in the absorption peak of –COO–. In conclusion, the *Salix psammophila* nanofibers presented heavy metal adsorption properties due to their small structural properties and the existence of carboxyl groups.

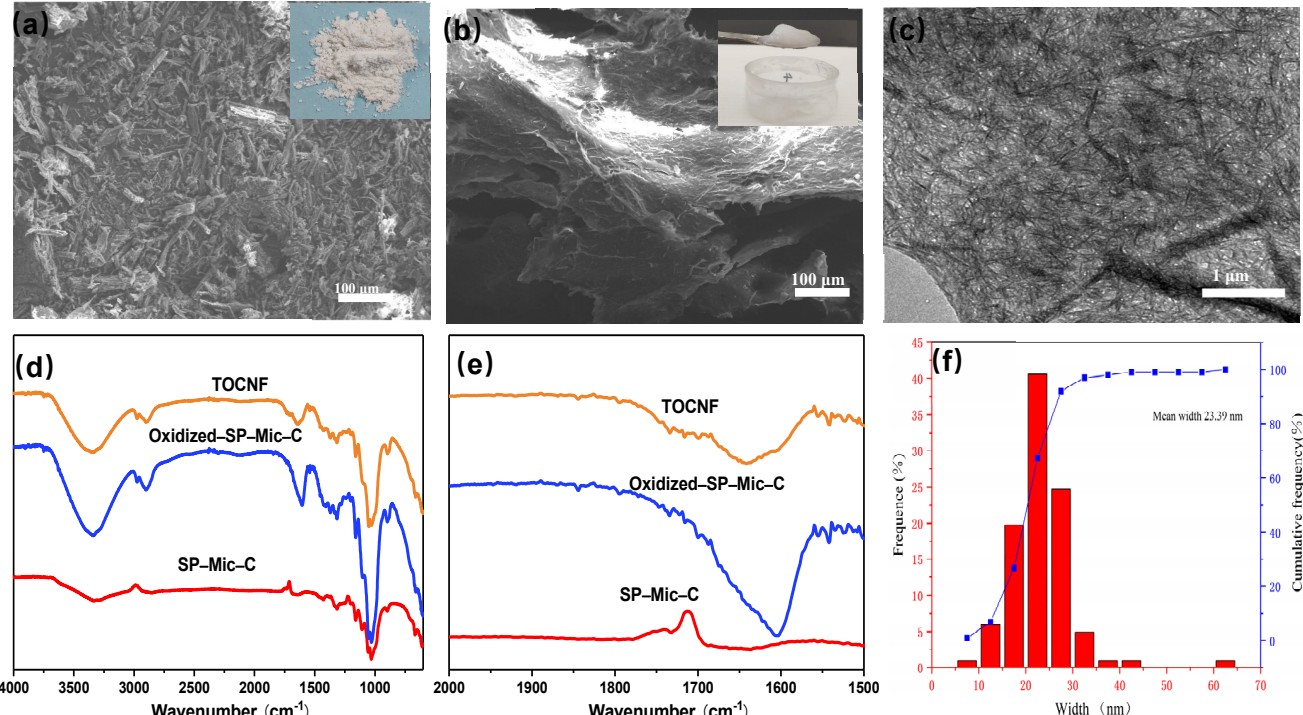

**Figure 1.** Scanning electron microscope (SEM) images of: (**a**) *Salix psammophila* microcrystalline cellulose (SP-Mic-C); (**b**) *Salix psammophila* nanofibers (TOCNF); (**c**) transmission electron microscope (TEM) images of TOCNF; (**d**,**e**) Fourier-transform infrared spectroscopy (FTIR) of SP-Mic-C, oxidized *Salix psammophila* microcrystalline cellulose (oxidized-SP-Mic-C), and TOCNF; (**f**) TOCNF diameter distribution.

### 3.2. Characterization of the Aerogel Spheres

#### 3.2.1. Scanning Electron Microscopy (SEM) of Aerogel Spheres

Figure 2 shows SEM images of sample aerogel spheres. The aerogel spheres had relatively complete spherical structures. From Figure 2, it can be seen that the samples showed a flaky 3D porous structure with relatively dense pores. The pure cellulose aerogel spheres had relatively uniform voids, and no filaments appeared on the surface of the pores. For aerogel spheres with the TOCNF addition, all the structural surfaces appeared

filament-like, and with the increase in TOCNF addition, the filament-covered surface became more obvious and the pore size gradually decreased. These results showed that the amount of active sites of the aerogel spheres increased with the addition of TOCNF while the addition of excess cellulose resulted in agglomeration, leading to decreases in the amount of active sites.

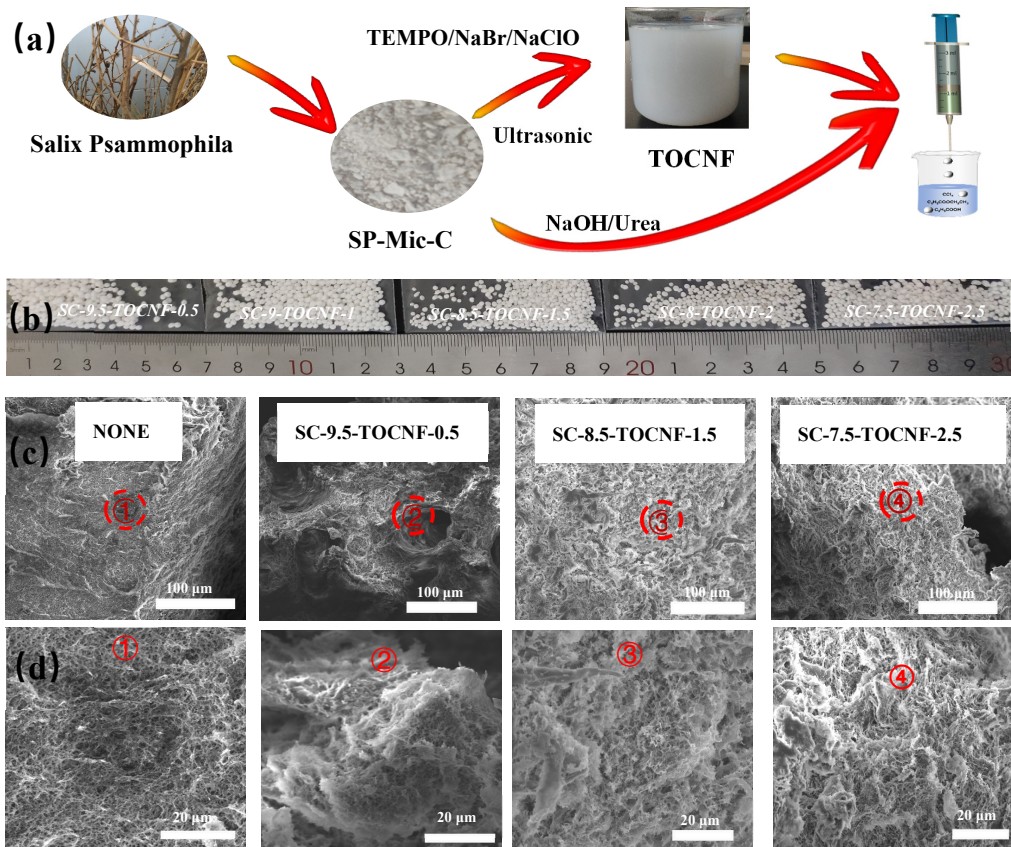

**Figure 2.** (**a**) Preparation flow chart of aerogel spheres; (**b**) images of sample aerogel spheres; (**c**,**d**) SEM images of aerogel spheres (samples were named as follows: *Salix psammophila* microcrystalline cellulose solution-mass of solution (g)-*Salix psammophila* nanofiber solution-mass of solution (g): ①: None; ②: SC-9.5-TOCNF-0.5; ③: SC-8.5-TOCNF-1.5; and ④: SC-7.5-TOCNF-2.5).

### 3.2.2. Fourier-Transform Infrared Spectroscopy (FTIR) of Aerogel Spheres

The infrared spectra of the samples are shown in Figure 3a,b. The peak of –CH$_3$ near 2900–2975 cm$^{-1}$ refers to –C–H stretching vibrations that were derived from methyl and methylene [31]. The stretching vibration peak of –C=O was near 1728 cm$^{-1}$. The asymmetric contraction vibration of carboxyl was near 1600 cm$^{-1}$, indicating that the TEMPO-treated samples had successfully introduced carboxyl functional groups and formed aerogel spheres after NaOH/urea dissolution because absorption peaks still existed. The results showed that the TOCNF did not change after dissolution, as shown in Figure 3b. The peak at 1424 cm$^{-1}$ could refer to the –C–H bending vibration peak of type I cellulose, indicating that the sample was always a cellulose type I crystalline structure. The peaks at 1050 and 897 cm$^{-1}$ are characteristic absorption peaks of cellulose that refer to the –C–O stretching vibrations [32] and –C–H oscillation vibrations of the cellulose, respectively. As shown in the FTIR spectra, the composite aerogel spheres possessed carboxyl groups, hydroxyl groups, and methoxy groups, all of which provided an effective complexation site for heavy metal adsorption.

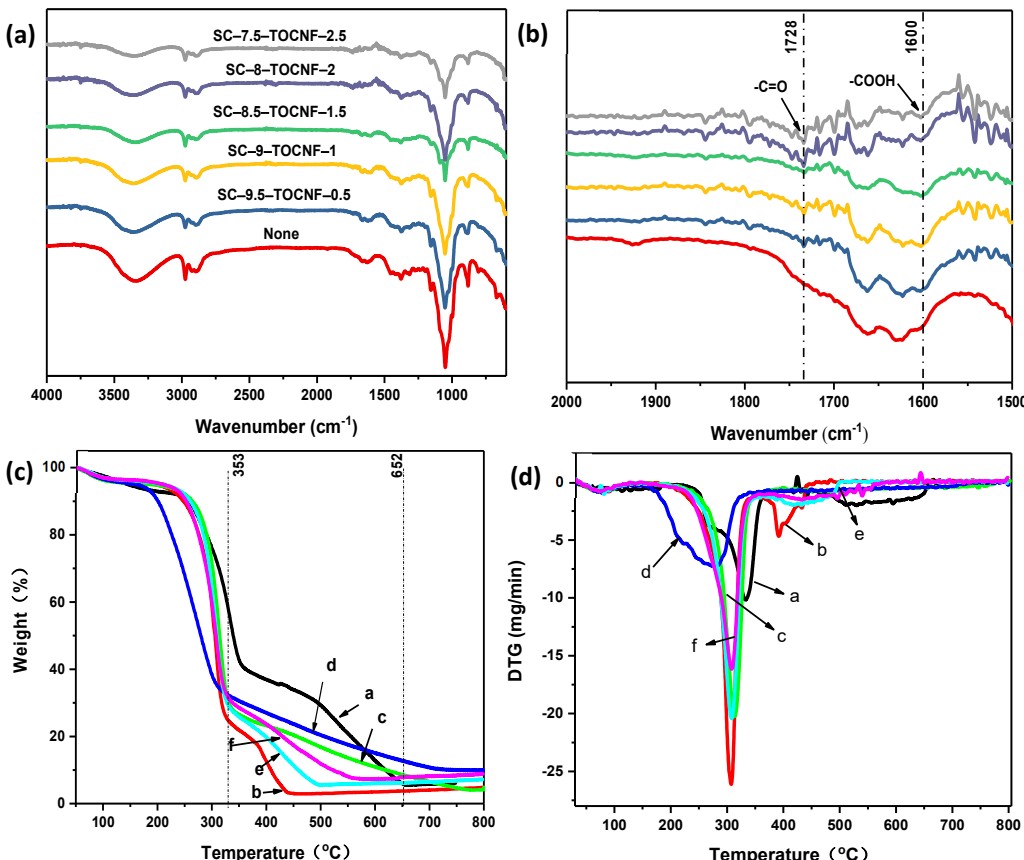

**Figure 3.** Fourier transform infrared analysis of aerogel spheres (**a**,**b**), (**c**) thermogravimetric analysis (TG) and (**d**) differential thermogravimetric analysis (DTG) results of aerogel spheres (the samples were named as follows: *Salix psammophila* microcrystalline cellulose solution-mass of solution (g)-*Salix psammophila* nanofiber solution-mass of solution (g): a: None; b: SC-9.5-TOCNF-0.5; c: SC-9.0-TOCNF-1.0; d: SC-8.5-TOCNF-1.5; e: SC-8.0-TOCNF-2.0; and f: SC-7.5-TOCNF-2.5.

3.2.3. Thermogravimetric Analysis (TGA) of Aerogel Spheres

Figure 3c,d shows the thermogravimetric analysis (TG) and differential thermogravi-metric analysis (DTG) results of the aerogel spheres. As can be seen from Figure 3c, all samples presented a weight loss of nearly 10% between 50 and 150 °C, indicating the volatilization stage of water. With the increase in temperature, the samples showed different pyrolysis curves. First, for blank aerogel spheres, the pyrolysis of cellulose created dehydrated sugars and alcohols between 200 and 350 °C. From 350 to 600 °C, the cellulose rapidly lost weight, mainly via small ketones, furans, alcohols, etc., and the re-pyrolysis of dehydrated sugars enabled the formation of smaller molecular compounds. Due to the addition of TOCNF, the thermal degradation temperature of the composite materials was advanced between 200 and 350 °C [33] because nanocellulose is a short cellulose molecular chain that is easily decomposed by heat, which further indicates that the nanocellulose was successfully doped into aerogel spheres. According to DTG analysis (Figure 3d), the maximum pyrolysis rate peak of the blank aerogel spheres was 333 °C. Compared to blank aerogel pellets, the thermal degradation of composite aerogel spheres occurred at least 20 °C lower. Obviously, the addition of nanocellulose reduced the thermal stability of the aerogel pellets to some extent. The thermal degradation peak at 400 °C is the thermal degradation absorption peak of residual compounds. In addition, the maximum thermal degradation rate of the composite aerogel spheres was higher than that of the blank aerogel spheres. The maximum thermal degradation rate of the composite aerogel spheres decreased with the increase in nanocellulose content. It has been proven that these aerogel spheres have good thermal stability.

### 3.2.4. The N$_2$ Adsorption–Desorption Isotherms (BET)

Figure 4 shows the adsorption–desorption curve and pore size distribution of the aerogel spheres. The aerogel spheres presented the same N$_2$ adsorption–desorption isotherm for both the type IV sorption isotherm and the H1 retention ring, so all samples had an abundant mesoporous structure [34]. Table 1 shows the BET specific surface, total pore volume, and average pore size of the composite aerogel spheres, demonstrating that the increase dosage of TOCNF led to decreases in the aperture size of the samples. The BET surface area of the blank aerogel spheres was 137.42 m$^2$·g$^{-1}$, the total pore volume was 0.8996 cm$^3$·g$^{-1}$, and the average pore size was 22.86 nm. For the composite aerogel spheres, SC-9-TOCNF-1 had the largest BET surface area and total pore volume. Compared to the blank aerogel spheres, the BET specific surface area of the doped TOCNF increased by 47.42 m$^2$·g$^{-1}$ and the total pore volume decreased by 0.0383 cm$^3$·g$^{-1}$. This was because the addition of the right amount TOCNF also played a supporting role in the pore structure, which increased the BET surface area of the samples while decreasing the total pore volume.

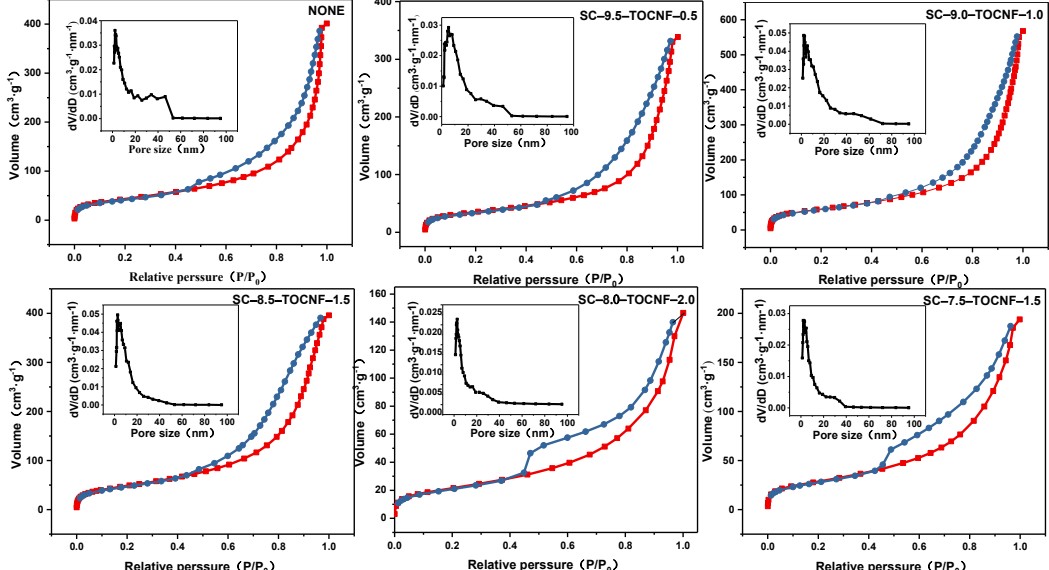

**Figure 4.** Brunner−Emmet−Teller measurements (BET) of aerogel spheres (samples were named as follows: *Salix psammophila* microcrystalline cellulose solution-mass of solution (g)-*Salix psammophila* nanofiber solution-mass of solution (g)).

**Table 1.** BET specific surface, total pore volume, and average pore size of composite aerogel spheres.

| Sample | BET Surface Area (m$^2$·g$^{-1}$) | Total Pore Volume (cm$^3$·g$^{-1}$) | Average Pore Size (nm) |
|---|---|---|---|
| NONE | 157.42 | 0.8996 | 22.860 |
| SC-9.5-TOCNF-0.5 | 115.90 | 0.5169 | 17.841 |
| SC-9.0-TOCNF-1.0 | 204.84 | 0.8613 | 16.819 |
| SC-8.5-TOCNF-1.5 | 171.76 | 0.6070 | 14.136 |
| SC-8.0-TOCNF-2.0 | 74.142 | 0.2175 | 11.736 |
| SC-7.5-TOCNF-2.5 | 100.11 | 0.2958 | 11.818 |

Notes: Samples were named as follows: *Salix psammophila* microcrystalline cellulose solution-mass of solution (g)-*Salix psammophila* nanofiber solution-mass of solution (g).

### 3.3. Heavy Metal Adsorption

Before the primary experimental work began, we carried out preliminary heavy metal adsorption tests on a series of samples (Table S1) with an adsorbent quality of 0.0500 g, and adsorption time of 120 min, an adsorption concentration of 500 mg/g, an adsorption temperature of 60 °C, and an adsorption pH of 6.5. The experimental results showed that the

adsorption capacity values of SC-10-TOCNF-0 (None) were 52.75, 78.63, and 96.65 mg/g for Cu(II), Mn(II), and Zn(II), respectively. Meanwhile, the values of the other aerogel spheres first increased and then decreased. Among them, SC-8.5-TOCNF-1.5 presented a good adsorption performance for three kinds of heavy metals, with adsorption capacity values of 267.64, 264.83, and 142.33 mg/g for Cu(II), Mn(II), and Zn(II), respectively. The adsorption capacity of SC-8.5-TOCNF-1.5 was significantly higher than that of None. Therefore, the SC-8.5-TOCNF-1.5 sample was used for subsequent experiments.

### 3.3.1. Effect of Initial Heavy Metal Concentration on Adsorption

The effect of the initial concentration of Cu(II), Mn(II), and Zn(II) on the adsorption capacity of the aerogel spheres is shown in Figure 5a. It can be seen that the trend of the adsorption capacity of the aerogel spheres to the three metal ions first rapidly increased and then kept stable near the initial concentration. This was because during the initial stage of adsorption, the adsorption process was mainly carried out on the aerogel spheres' surfaces. With the increase in time, the adsorption process gradually shifted to the inner channel of the adsorbent. When the concentration was further increased, the adsorption capacity remained stable due to the saturation of the active adsorption site. Therefore, Cu(II) with an initial concentration of 400 mg/L, Mn(II) with an initial concentration of 300 mg/L, and Zn(II) with an initial concentration of 300 mg/L were selected as the ideal initial concentration conditions.

### 3.3.2. Effect of Temperature on Adsorption

The relationship between adsorption capacity and adsorption temperature is shown in Figure 5b. With the increase in solution temperature, the adsorption capacity of the aerogel spheres to copper and zinc ions first increased and then decreased. The reason for this is that with the increase in the adsorption temperature of the system, the hydrogen bonds between the macromolecular chains of the cellulose structure were weakened and the number and the adsorption capacity of the activated adsorption sites were correspondingly enhanced, which was conducive to the adsorption between the heavy metal ions and the aerogel spheres. However, if the temperature continued to rise because the adsorption process was exothermic, it would have inevitably led to the weakening of adsorption and the reduction of adsorption capacity. Therefore, the optimum adsorption temperatures of aerogel spheres for Cu(II), Zn(II), and Mn(II) were 60, 50, and 70 °C, respectively.

### 3.3.3. Effect of pH Values on Adsorption

The adsorption rate of the aerogel spheres onto Cu(II), Mn(II), and Zn(II) varied with pH, as shown in Figure 5c, which shows that pH had an obvious effect on the adsorption performance of the material, with inconsistent changes. The adsorption of Cu(II) onto the aerogel spheres increased with the increase in pH, and the aerogel spheres had the best adsorption capacity for Cu(II) when the pH was 6. However, Mn(II) and Zn(II) were quite different from Cu(II); their pH first increased with the increase in pH and then decreased when the pH was greater than 4. It is speculated that they were affected by the increase in pH, resulting in a hydrolysis reaction that influenced the adsorption effect. Therefore, the optimal adsorption pH of Mn(II) and Zn(II) was 4.

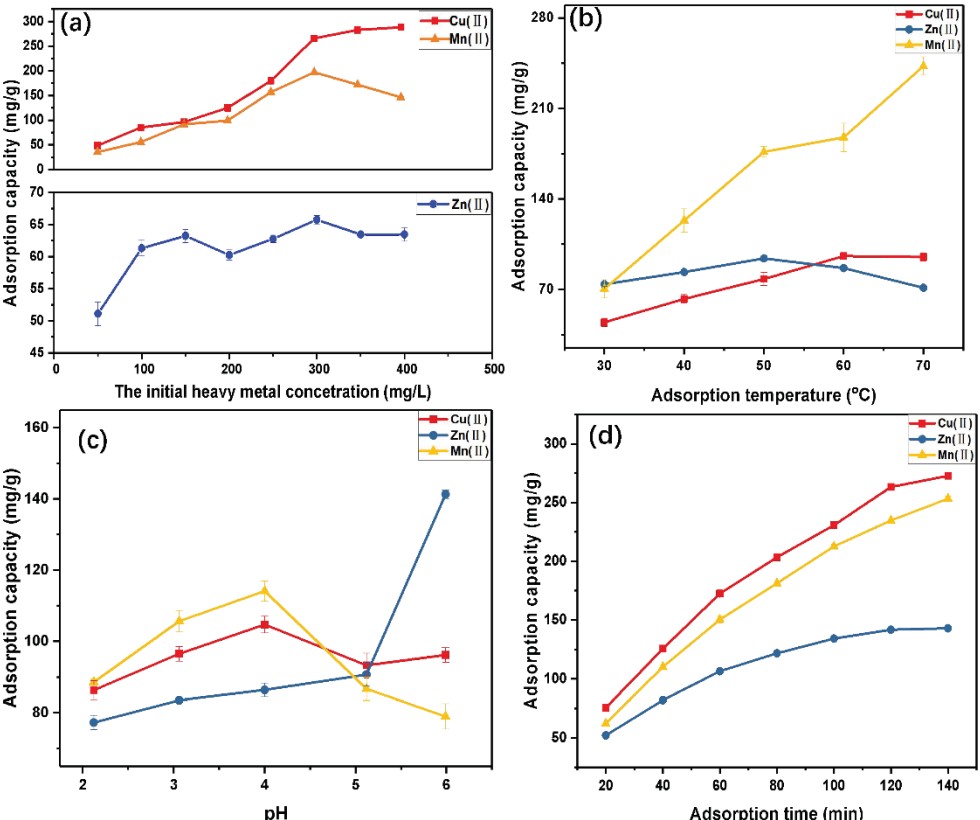

**Figure 5.** Effect of (**a**) the initial heavy metal ion (adsorbent quality: 0.0500 g; pH: 6.0; temperature: 60 °C; time: 120 min), (**b**) adsorption temperature (adsorbent quality: 0.0500 g; pH: 6.0; solution concentrations of 400 mg/g for Cu(II), 300 mg/g for Mn(II), and 300 mg/g for Zn(II); time: 120 min); (**c**) pH (adsorbent quality: 0.0500 g; temperatures for 60 °C for Cu(II), 70 °C for Mn(II), and 50 °C for Zn(II); solution concentrations of 400 mg/g for Cu(II), 300 mg/g for Mn(II), and 300 mg/g for Zn(II); time: 120 min); (**d**) adsorption time (adsorbent quality: 0.0500 g; temperatures of 60 °C for Cu(II), 70 °C for Mn(II), and 50 °C for Zn(II); solution concentrations of 400 mg/g for Cu(II), 300 mg/g for Mn(II), and 300 mg/g for Zn(II); pH values of 4.0 for Cu(II), 4.0 for Mn(II), and 6.0 for Zn(II)) of SC-8.5-TOCNF-1.5 aerogel spheres.

### 3.3.4. Effect of Time on Adsorption

The adsorption amounts of heavy metals at different times on the aerogel spheres are shown in Figure 5d. With the increase in time, adsorption gradually reached saturation. Within two hours, the adsorption amounts of Cu(II), Mn(II), and Zn(II) were 263.26, 234.73, and 141.83 mg/g, respectively. The aerogel spheres had the best adsorption effect on the three kinds of Cu(II), followed by Mn(II). The results showed that the aerogel spheres had good adsorption properties for Cu(II) and Mn(II), but their adsorption capacity for Zn(II) was low.

### 3.4. Kinetic Studies

The adsorption kinetics results are tabulated in Table 2, and the fitting models are shown in Figure 6. In accordance with the results presented in Figure 6 and Table 2, which enable comparisons of the experimental equilibrium adsorption capacities for the adsorption of three heavy ions onto the aerogel spheres, Figure 6a shows a plot of log $(q_e - q_t)$ as a function of time according to the linearized pseudo-first-order model. The low $R^2$ values for Cu(II), Mn(II), and Zn(II) of 0.8612, 0.9484, and 0.8694, respectively, show that the adsorption of the three heavy ions did not follow the pseudo-first-order model. However, Figure 6b presents the plot of $(t/q_t)$ versus t in the linearized pseudo-second-order model, which presents an excellent linear fit. The adsorption kinetics $R^2$ values for Cu(II), Mn(II),

and Zn(II) of 0.9895, 0.9980, and 0.9986, respectively, indicate that the adsorption process of the aerogel spheres for three heavy metal ions was chemisorption. The carboxyl groups and hydroxyl groups of the aerogel spheres provided a large number of active sites for heavy metal adsorption.

**Table 2.** Kinetic parameters for Cu(II), Mn(II), and Zn(II) ion adsorption onto aerogel spheres.

| Metal | Pseudo-First-Order Kinetic Model | | Pseudo-Secondary-Order Kinetic Model | |
|---|---|---|---|---|
| | $K_1$ (min$^{-1}$) | $R^2$ | $K_2$ (min$^{-1}$) | $R^2$ |
| Cu(II) | 0.0636 | 0.8612 | 0.00006 | 0.9895 |
| Zn(II) | 0.0925 | 0.8694 | 0.00017 | 0.9986 |
| Mn(II) | 0.0520 | 0.9484 | 0.00006 | 0.9980 |

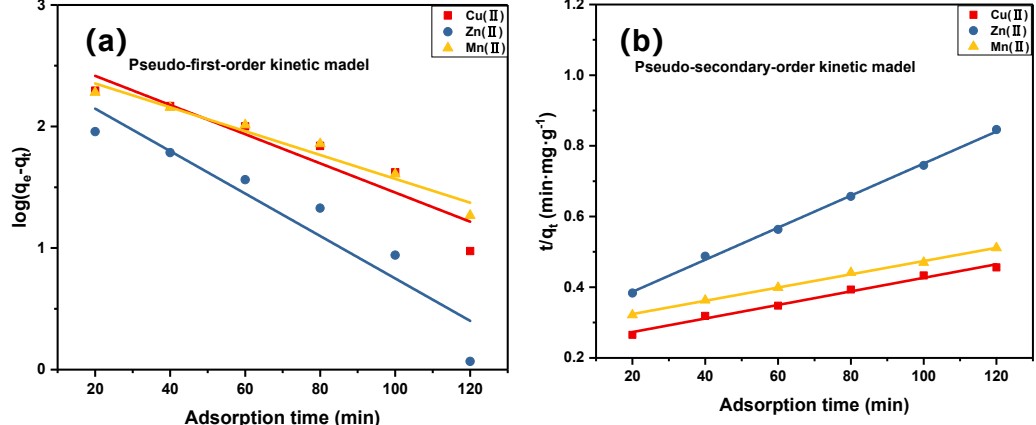

**Figure 6.** (**a**) Pseudo-first-order kinetic model and (**b**) pseudo-secondary-order kinetic model fitting curves of the experimental data at optimum reaction conditions.

### 3.5. FTIR following Heavy Metal Adsorption

The FTIR results following Cu(II), Mn(II), and Zn(II) adsorption are shown in Figure 7, which are in contrast to those shown in Figure 3b; after the adsorption, the stretching vibration peak of –C=O and the asymmetric contraction vibration of carboxyl near 1600 cm$^{-1}$ were weakened to some extent, indicating that the functional groups of the aerogel ball surfaces combined with heavy metal ions. The experimental results show that the adsorption of heavy metals was mainly chemical adsorption.

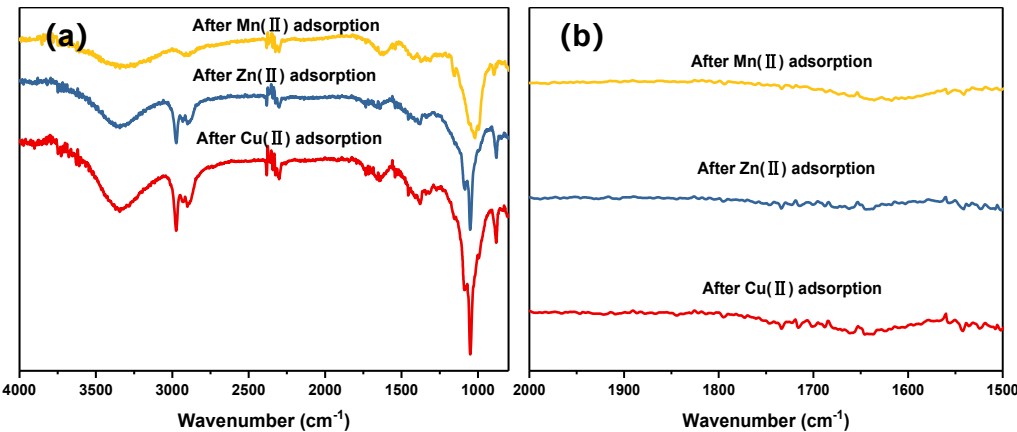

**Figure 7.** FTIR results of Zn(II), Mn(II), and Cu(II) adsorption. (**a**): The range of wavenumber is 4000–600 cm$^{-1}$; (**b**): The range of wavenumber is 4000–600 cm$^{-1}$.

### 3.6. Comparison with Previously Reported Data for Cu(II), Mn(II), and Zn(II)

Table 3 compares the data of aerogel spheres with other heavy metal ion adsorbents. It can be seen from the table that *Salix psammophila* microcrystalline cellulose/*Salix psammophila* nanofiber aerogel spheres demonstrated good adsorption for Cu(II) and Mn(II). It can be concluded that this material can be widely used for heavy metal adsorption.

**Table 3.** Comparison of adsorption properties of different materials for $Cu^{2+}$, $Mn^{2+}$, and $Zn^{2+}$.

| Sample | Cu(II) (mg/g) | Mn(II) (mg/g) | Zn(II) (mg/g) | Author |
|---|---|---|---|---|
| This work | 272.69 | 253.25 | 143.00 | This work |
| C6 carboxylic microcrystalline cellulose | 165.5 0 | — | — | Jifeng L [35] |
| Chinar cellulose-graft-poly amidoxime | 84.00 | — | — | Panpan J [36] |
| Polyethyleneimine functionalized cellulose nanofiber magnetic composites | 93.71 | —·- | — | Guo Z [37] |
| Carboxymethylated CNFs | 115.3 | — | — | Qin F [38] |
| Core-shell magnetic rosin-based polymer $Fe_3O_4$@ RPM microspheres | — | 45.00 | — | Kechun L [39] |
| $Fe_3O_4$ nanoparticles | — | 36.81 | — | Liu Y [40] |
| Sulfhydryl-modified cassava straw | — | — | 60.24 | Deng H [41] |
| Lignocellulose@ activated clay nanocomposite | — | — | 315.90 | Zhang X [42] |

### 4. Conclusions and Perspectives

In summary, we modified *Salix psammophila* microcrystalline cellulose (SP-Mic-C) and prepared the *Salix psammophila* nanofibers (TOCNF) with high-intensity ultrasound, and then we mixed the solution of SP-Mic-C and TOCNF to prepare aerogel spheres derived from *Salix psammophila* (ASSP). The experimental conclusions are as follows. (1) TOCNF is a whisker structure with an average diameter of 23.39 nm, and the appearance of a peak near 1600 $cm^{-1}$ in Fourier-transform infrared spectroscopy indicates the existence of carboxyl functional groups. (2) The porous structure of ASSP and the introduction of TOCNF enable materials to have excellent adsorption properties for heavy metals. ASSP was shown to have maximum adsorption capacities on Cu(II), Mn(II), and Zn(II) of 272.69, 253.25, and 143.00 mg/g, respectively, and the adsorption process fit the pseudo-secondary-order kinetic model and was chemical adsorption. This work indicates that the successful preparation of ASSP is a promising way to utilize abandoned agricultural and forestry resources and protect the ecological environment, as well as providing a reference for researchers to study abandoned agricultural and forestry resources. In the future, ASSP can be used as a carrier to load photocatalyst and by widely used in the photocatalytic degradation of dyes and volatile organic compounds (VOCs).

**Supplementary Materials:** The following supporting information can be downloaded at: https: //www.mdpi.com/article/10.3390/f13010061/s1, Figure S1: Drawing of heavy metal standard curve. Table S1: Drawing of heavy metal standard curve (adsorbent quality: 0.0500 g; adsorption time: 120 min; adsorption Concentration: 500 mg/g; adsorption Temperature: 60 °C; adsorption pH: 6.5).

**Author Contributions:** Writing—original draft Y.Z.; investigation Y.A. and B.W.; methodology, Z.C. and K.W.; formal analysis, W.Z.; data curation, X.W. (Xiao Wang) and Z.H.; writing—review and editing, X.Z. and X.L.; resources, S.W.; supervision, X.W. (Ximing Wang). All authors have read and agreed to the published version of the manuscript.

**Funding:** This research was funded by Natural Science Foundation of Inner Mongolia, grant number 2020MS02007, the High-Level Talent Research Initiation Project of Inner Mongolia Agricultural University, grant number NDYB2018-59, Inner Mongolia Autonomous Region Postgraduate Research and Innovation Funding Project BZ2020055, Science and technology innovation fund for college students of Inner Mongolia Agricultural University (KJCX2021038), Science and Technology Achievements Transformation Project of Inner Mongolia Autonomous Region in China, grant number CGZH2018136, and Grass Talents Engineering Youth Innovation and Entrepreneurship of Inner Mongolia Autonomous Region in China grant number Q2017053. Inner Mongolia Autonomous

Region Postgraduate Research and Innovation Funding Project:BZ2020055; Science and Technology Innovation Fund for College Students of Inner Mongolia Agricultural University: KJCX2021038.

**Data Availability Statement:** Not applicable.

**Conflicts of Interest:** The authors declare that there is no conflict of interests regarding the publication of this paper.

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
