# Peer review of "Evaluation of Aerogel Spheres Derived from Salix psammophila in Removal of Heavy Metal Ions in Aqueous Solution"

_forests, doi:10.3390/f13010061_

Round 1

Reviewer 1 Report

The current study entitled “Evaluation of Aerogel Spheres derived from Salix Psammophila on Removal of Heavy Metal Ions in Aqueous Solution” is good. For a better understanding in-depth, it is a need for time to work on this topic. Furthermore, the achievement of potential benefits by using current technology is also dependent on the extensive research work for more exploration. Although the experiment is well organized, yet I suggest a rejection due to the following deficiencies.

Major Concerns

  • Systematic abstract is missing. Introduce the need for study in 1-2 lines.
  • Please give a clear cut point problem source as a problem statement that is tackled in the current study.
  • Give logical reason for the selection of current strategy i.e., Aerogel Spheres derived from Salix Psammophila.
  • Quantitative data is also important to support your conclusion. Would you please provide some quantitative data in terms of percentage significant increase or decrease in the abstract?
  • Please provide a conclusive conclusion with is withdrawn through research in a single line. The statement “The adsorption kinetics shows that the adsorption process accorded with the Pseudo-secondary order kinetic model” is general. Please conclude with a statement that shows a knowledge gap covered, potential beneficiaries and specific recommendations as well.
  • Give future prospective in a single line.
  • As per standard suggestions, please avoid using title words as keywords
  • Please follow the title in the introduction section, i.e., Aerogel Spheres derived from Salix Psammophila, then Removal of Heavy Metal Ions in Aqueous Solution, knowledge gap, hypothesis and aims.
  • Also, provide a novelty statement at the end. What new things authors have done or correlated in this research compared to old ones?
  • Would you please give a single line about the knowledge gap which your research has covered along with the hypothesis statement?
  • Material and methods are ok.
  • In results, each table must be self-explanatory. Please provide the abbreviation details in the end of each table and figure.
  • Please give a conclusive conclusion.
  • If the authors are not sure, then give future recommendations for more research and investigation.
  • Add the targeted beneficiary audience who will get benefits from this research.
  • Also, give clear-cut recommendations and future prospective regarding this research.

Reviewer 2 Report

The manuscript "Evaluation of Aerogel Spheres derived from Salix Psammophila on Removal of Heavy Metal Ions in Aqueous Solution" by 
Yuan Zhong et al. reports on the preparation of a novel prospective absrobent for the removal of heavy metal ions from industrial wastes based on nanostructured cellulose obtained from a replenishable resource, such as desert plant called Salix Psammophila. The manuscript's topic is urgent and falls well into the scope covered by  Forests. The manuscript is generally well structured and written and conslusions are meaningful.

The manuscript can be accepted for publication subject to a few minor corrections:

1) The first (lines 33-40) and second (lines 41-48) are identical. One should be removed

2) The authors should indicate the type and commercial provider of the heavy metal salts used for the absorption experiments. Does the absorption capability depend on the counter-ion of the heavy-metal cation (chloride, nitrate, carboxylate)? Were manganese compounds in the oxidation states other than Mn(II) tested?

3) Did the authors use only artificially prepared solutions of heavy-metal ions? What about real industrial wastewater probes?

4) What was the exact procedure to measure diameter distribution of TOCNF samples as shown in Fig. 5,f?

5) Please, add a bried description on the regeneration strategy of spent absorbents. Could the heavy metals be desorbed from the nanocellulose absorbents by a specific chemical treatment for their repetitive usage? Should they be combusted to recover metal-containing species? 

Round 2

Reviewer 1 Report

The author has modified the manuscript as per my suggestion. I suggest accepting the manuscript in its present form.

This manuscript is a resubmission of an earlier submission. The following is a list of the peer review reports and author responses from that submission.